# An Evaluation of the Crop Preference and Phenotypic Characteristics of *Ceracris kiangsu* Tsai (Orthoptera: Arcypteridae) under Different Temperatures

**DOI:** 10.3390/biology12111377

**Published:** 2023-10-27

**Authors:** Meizhi Wang, Hongmei Li, Abdul Aziz Bukero, Jinping Shu, Fuyan Zhuo, Linyi Liu, Aihuan Zhang

**Affiliations:** 1MARA-CABI Joint Laboratory for Bio-Safety, Institute of Plant Protection, Chinese Academy of Agricultural Science, Beijing 100193, China; wmzwmza@163.com (M.W.); a.bukero@cabi.org (A.A.B.); 2College of Bioscience and Resource Environment, Beijing University of Agriculture, Beijing 102206, China; zhangaihuan@126.com; 3CABI East and Southeast Asia, Beijing 100081, China; 4Research Institute of Subtropical Forestry, Chinese Academy of Forestry, Hangzhou 311400, China; shu_jinping001@163.com; 5National Agro-Tech Extension and Service Center, Beijing 100125, China; zhuofuyan@agri.gov.cn; 6State Key Laboratory of Remote Sensing Science, Aerospace Information Research Institute, Chinese Academy of Sciences, Beijing 100094, China; liuly01@aircas.ac.cn

**Keywords:** yellow-spined bamboo locust, invasion, staple crop, biology, phenotypic traits

## Abstract

**Simple Summary:**

The invasion of the yellow-spined bamboo locust (YSBL) has spread widely from Southeast Asian countries to Southwest China in recent years, leading to potential production losses. Understanding the adaptability of the YSBL to different hosts and temperatures is essential in facilitating early warning monitoring and detection. This study aimed to discover the crop host preference of YSBL nymphs via life table and fitness characteristics. The results showed that the YSBL may be harmful to wheat, rice, and maize. A temperature of 30 °C is ideal for the development and survival of YSBL populations. Under laboratory conditions, YSBL nymphs prefer the seedlings of wheat and rice to maize. The current findings will provide the fundamental information needed to develop a management strategy.

**Abstract:**

The yellow-spined bamboo locust (YSBL), *Ceracris kiangsu* Tsai, has historically had a significant impact on different bamboo varieties in East Asia and Southeast Asia. Since 2014, there have been many outbreaks of YSBL populations in Laos, and YSBLs subsequently invaded Southwest China in 2020 and 2023. However, there was limited information about the damage to staple crops. Life table parameters and fitness parameters were assessed using wheat, rice, waxy maize, and sweet maize under three different temperatures (25 °C, 30 °C, and 35 °C) in the laboratory. The results indicated that the YSBLs feeding on wheat seedlings displayed a significantly higher survival rate, a shorter developmental time, and a higher adult emergence rate compared to YSBLs feeding on the other host species at 30 °C. The developmental durations of 1st and 3rd instar YSBLs on wheat (1st: 8.21 ± 0.35 d; 3rd: 6.32 ± 0.34 d) and rice (1st: 7.19 ± 0.23 d; 3rd: 9.00 ± 0.66 d) were significantly shorter than those of 1st and 3rd instar YSBLs on waxy maize (1st: 13.62 ± 1.22 d; 3rd: 13.67 ± 6.33 d) and sweet maize (1st: 16.00 ± 1.79 d; 3rd: 18.00 ± 3.49 d) at 30 °C. The body lengths of male and female YSBLs on wheat (male: 29.52 ± 0.40 mm, female: 34.97 ± 0.45 mm) and rice (male: 28.85 ± 0.68 mm, female: 34.66 ± 0.35 mm) were significantly longer than those observed when they were fed on sweet maize (male: 25.64 ± 1.60 mm, female: 21.93 ± 6.89 mm). There were only male adults obtained on waxy maize. The phenotypic characteristics of the YSBLs feeding on rice seedlings were very close to those of the YSBLs feeding on wheat seedlings. A relatively slower decline was observed in the survival rates of YSBL nymphs on wheat and rice compared to those on waxy maize and sweet maize at 25 °C, 30 °C, and 35 °C. In short, this study implied that YSBLs prefer wheat and rice. This study is the first report of direct damage caused by the YSBL to wheat in the laboratory, and its results could be useful in improving our understanding of the host preference of the YSBL and providing strategies for the management of this pest in field crops.

## 1. Introduction

For phytophagous insects, plant hosts represent a fundamental requisite that supports their survival and that may have extensive effects on the fitness, fecundity, and reproductive strategies of these insects [1,2]. The adult longevity of *Monolepta hieroglyphica* (Motschulsky) fed on maize filament was significantly longer than when it was fed on potato leaves, maize leaves, sunflower leaves, and rice leaves [3]. *Spodoptera frugiperda* (J.E. Smith) larvae fed on rice had a significantly longer developmental time than those fed on maize [4]. Significantly for migratory or invasive pests, they may cause damage to new plants after they become established in new areas, such as, for example, *Bactrocera zonata* (Saunders) developing in *Olea europaea* L. after the first report in Egypt [5].

The yellow-spined bamboo locust (YSBL), *Ceracris kiangsu* Tasi (Orthoptera: Arcypteridae) has been recorded in East Asian and Southeast Asian countries such as China, Laos, and Vietnam. In 2020, a migration of YSBLs occurred in large numbers from Laos and Vietnam to China [6,7]. During this migration, YSBLs were reported to have entered agricultural areas in Yunnan Province, China and caused damage to rice, sugarcane, and maize [8,9]. About 200 ha of forest land were invaded in Yunnan Province, China by YSBLs from Laos in 2023, and the swarms of invasive YSBL adults fed on bamboo leaves, plantain, and tiger grass [10]. However, limited information has been reported on the range of YSBL damage to important staple crops and the life cycle of this pest on different crops. Globally, wheat (*Triticum aestivum* L.), rice (*Oryza sativa* L.), and maize (*Zea mays* L.) are the top three key crops, all belonging to the Poaceae family [11]. These three crops are widely grown in China from the south to the north, and the temperature in different growing regions for the same crop varies greatly [12]. Furthermore, worldwide, more than 250 million people depend on wheat as a staple food [13].

Temperature is the most basic factor in the development of insects, directly affecting their development rate and population dynamics [14,15]. Therefore, the effects of temperature and staple crops on the YSBL should be investigated. Understanding the effects of temperature and host plants on migratory pests is crucial for agroforestry protection. Outbreaks of migratory pests may have pressing economic and food security implications [16]. Only a few reports have addressed these implications, detailing YSBL damage to rice and maize in the field [17,18]. The life table and its parameters, such as developmental duration, age-specific survival rate, and age-stage specific life expectancy, can be used to better understand and analyze insect population dynamics, make predictions, and develop integrated pest management plans [19]. The body length of *Aeolothrips intermedius* Bagnall was the phenotype characteristic that was most responsive to the host plant [20]. The body weight of *Hyphantria cunea* (Drury) feeding on high-preference host plants was significantly higher than that of those feeding on low-preference host plants [21]. In this study, the survival and development of nymphs and the morphometrics of the YSBL were examined using one of the following plants: seedlings of common wheat, rice, waxy maize (*Z. mays* L. *sinensis* Kulesh), and sweet maize (*Z. mays* L. *var. rugosa* Bonaf.). They were evaluated at 25 °C, 30 °C, and 35 °C. This evaluation will contribute to our understanding of the effect of agricultural host plants on YSBL development. The results will help to further advance phenology models for the YSBL based on life-cycle information, and they will provide basic information about potential distribution in the agricultural region.

## 2. Materials and Methods

### 2.1. Insect Culture 

The egg pods of YSBLs were initially collected in mid-March 2022 from their natural bamboo habitat, Anhua County, Hunan Province, China (28°62′ N, 111°35′ E). The egg pods were incubated in a chamber (MGC-1000HP-2, Shanghai Yiheng Scientific Instrument Co., Ltd., Shanghai, China) at 30 ± 1 °C and 65 ± 5% relative humidity (RH) according to Fang et al. [22]. 

Once hatching occurred, 20 nymphs with the same hatching date were transferred gently into one plastic cylinder (17.8 cm height × 9 cm diameter). Different host seedlings were provided at the same time, and every cylinder was covered with gauze for air ventilation. The cylinders were transferred to different conditions for further tests.

For this study, we chose four different Chinese staple crops, including common wheat (Aikang 58, Henan Bainong Seed Industry Co., Ltd., Xinxiang, China), rice (Xiangzaoxian 45, Hunan Dongting Gaoke Seed Industry Co., Ltd., Yueyang, China), waxy maize (Jinghuangnuo, Beijing Huaao Nongke Jade Breeding Development Co., Ltd., Beijing, China), and sweet maize (Suketian 1506, Nanjing Jiahua Agricultural Development Co., Ltd., Nanjing, China), which were disease-resistant or widely cultivated in China [23,24,25,26]. The soaked seeds were planted with wet nutrient soil in the plastic pots (20 cm height × 20 cm diameter), covered with preservative film to preserve their humidity, and watered appropriately. Fresh healthy seedlings about 18 cm high were used to feed the insects. The seedlings with soil were removed from the plastic pots and wrapped with preservative film to maintain the soil moisture for fresh seedlings.

All treatments were carried out at three temperature regimes (25 °C, 30 °C, and 35 °C), at 70 ± 5% relative humidity (RH), and in 16:8 light (L): dark (D). The feces were cleaned away every 2 d to prevent microbial contamination of the food sources.

### 2.2. Experimental Setup of Life Table

A total of 60 newly hatched nymphs were prepared on every host plant at every temperature inside three plastic cylinders. The number of dead and alive individuals as well as the developing instars of the YSBL in every plastic cylinder were counted and recorded daily, and the sex of every adult was recorded after the nymphs emerged. Extra nymphs were placed in the cylinder if there were five individuals dead daily at the 1st instar, which would guarantee 20 individual nymphs in the 2nd instar.

The body length and body weight of instars from the 2nd to the 5th and adults were measured using electronic vernier calipers (0–150 mm, Shanghai Shenhan Measuring Tools Co., Ltd., Shanghai, China) and an electronic balance measurement (PL203, Mettler-Toledo Instruments Shanghai Co., Ltd., Shanghai, China), respectively. 

### 2.3. Data Analysis

The mean values of population parameters and phenotype characteristics were compared using t-tests and performing one-way ANOVA using SPSS 22.0 (IBM Corp., Armonk, NY, USA). The regression models and correlation relationship were analyzed using SPSS 22.0. The age-specific survival rate (*l_x_*) and age-stage life expectancy (*e_xj_*) were calculated according to Chi et al. [27] using the program TWOSEX-MSChart [28].

## 3. Results

### 3.1. Development of YSBL Nymphs 

The developmental durations of the YSBL nymphs feeding on different hosts at the same temperatures were compared (Figure 1). The YSBL nymphs feeding on the seedlings of wheat and rice were able to complete the entire nymphal stage at 25 °C, 30 °C, and 35 °C, while the YSBL nymphs only developed up to the 4th instar at 25 °C feeding on the seedlings of waxy maize, and at 30 °C and 35 °C feeding on the sweet maize (Figure 1). The developmental durations of YSBL nymphs feeding on the seedlings of wheat and rice were significantly shorter than the developmental durations of those feeding on the seedlings of waxy maize and sweet maize (25 °C: wheat 64.10 ± 2.23 d versus rice 62.60 ± 1.96 d versus sweet maize 86.00 ± 0.00 d; 30 °C: wheat 45.13 ± 2.40 d versus rice 50.27 ± 1.77 d versus waxy maize 60.00 d; 35 °C: wheat 39.00 ± 2.12 versus rice 47.00 d versus waxy maize 71.00 d) (*p* < 0.05). The above-mentioned variation trends were similarly found in the 1st instar, 2nd instar, 3rd instar, and 4th instar nymphs (at 25 and 35 °C), and in the 5th instar nymphs (at 25 °C).

Analysis indicated that there was a decreasing tendency in the developmental duration of the YSBL nymphs with increasing temperature (wheat: 25 °C 64.10 ± 2.23 d versus 30 °C 45.13 ± 2.40 d versus 35 °C 39.00 ± 2.12 d; rice: 25 °C 62.60 ± 1.96 d versus 30 °C 50.27 ± 1.77 d versus 35 °C 47.00 d; waxy maize: 30 °C 60.00 d versus 35 °C 71.00 d; sweet maize: 25 °C 86.00 ± 0.00 d) (*p* < 0.05) (Figure 1). The developmental durations of the 1st instar (on wheat, rice, waxy maize, and sweet maize) and the 2nd to the 4th instars (on wheat and rice) at 25 °C were significantly longer than the developmental durations of those instars at 35 °C (*p* < 0.05), while the developmental durations of 5th instar nymphs feeding on seedlings of wheat and rice at 25 °C were significantly longer than the developmental durations of those nymphs at 30 °C (*p* < 0.05).

### 3.2. Relationship between Temperature and Growth Rate of YSBL Nymphs

Regression quadratic models relating YSBL nymphal developmental rates and temperatures were established for different treatments. The developmental rates of YSBLs fed on wheat and rice seedlings significantly correlated with temperatures at early (N1–N2) and mature (N5) nymphal stages (Table 1). The other conditions could not be modeled as there was no significant difference.

### 3.3. Adult Emergence of YSBLs

The adult emergence rate of YSBLs fed on the wheat seedlings was the highest and was higher than that of YSBLs fed on the other three crops at 25 °C, 30 °C, and 35 °C (Table 2). The adult emergence rate of YSBLs fed on wheat seedlings was significantly higher than that of YSBLs fed on waxy maize at 30 °C (*p* < 0.05).

The highest adult emergence rates of YSBLs fed on wheat and rice seedlings were observed at 30 °C; these were higher than at 25 °C and 35 °C (Table 2). The adult emergence rate of YSBLs fed on rice at 30 °C was significantly higher than that of YSBLs fed on rice at 35 °C (*p* < 0.05).

### 3.4. Age-Specific Survival Rates and Life Expectancy of YSBLs

According to age-specific survival rate (*l_x_*) curves, a relatively slower decline was observed in the preadult survival rates of YSBLs on wheat and rice compared to waxy maize and sweet maize at 25 °C, 30 °C, and 35 °C, respectively (Figure 2). YSBLs fed on the seedlings of wheat and rice developed faster than those fed on waxy maize and sweet maize (wheat: 57 d, 34 d and 35 d at 25 °C, 30 °C, 35 °C; rice: 59 d, 42 d, 47 d at 25 °C, 30 °C, 35 °C, respectively; waxy maize: 60 d and 71 d at 30 °C, 35 °C, respectively; sweet maize: 86 d at 35 °C).

The first age (FA) means that the survival rate of the day was not more than 0.5 (i.e., *l_x_* ≤ 0.5). The FA at 25 °C occurred at 14 d and 13 d on wheat and rice, respectively, which was 7–9 days later than for those reared on waxy maize and sweet maize. Similarly, the FA at 30 °C occurred at 9 d and 8 d on wheat and rice, which was 3–5 days later than those reared on the other two host plants, while the FA at 35 °C occurred at 8 d on wheat and waxy maize, which was 2–4 days later than for those reared on the other two host plants (Figure 2). 

Under the same host, the age-stage specific life expectancy (*e_xj_*) of YSBLs shortened with increasing temperature, showing a decreasing trend (Figure 3). At the same temperature, the *e_xj_* of YSBLs fed on wheat and rice was longer than when they were fed on sweet corn and waxy corn (Figure 3). 

### 3.5. Phenotype Characteristics of the YSBL and Its Relationship with Different Hosts

On the same host, the bodies of YSBL nymphs lengthened as the instar increased (Figure 4). The body length of female adults fed on wheat and rice seedlings was significantly longer than that of female adults fed on sweet maize (wheat: 34.97 ± 0.45 mm versus rice: 34.66 ± 0.35 mm versus sweet maize: 21.93 ± 6.89 mm) (*p* < 0.05). There were no female adults obtained on waxy maize. The body length of male adults fed on wheat and rice seedlings was significantly longer than that of those fed on waxy maize and sweet maize (wheat: 29.52 ± 0.40 mm versus rice: 28.85 ± 0.68 mm versus waxy maize: 25.11 ± 2.62 mm versus sweet maize: 25.64 ± 1.60 mm) (*p* < 0.05).

Similarly, the body weight of YSBL nymphs was heavier as the instar increased. YSBL male adults that fed on wheat and rice seedlings weighed significantly more than those fed on waxy maize, but not significantly more compared with sweet maize (Wheat: 0.228 ± 0.014 g versus rice: 0.265 ± 0.023 g versus waxy maize: 0.137 ± 0.038 g versus sweet maize: 0.212 ± 0.019 g) (*p* < 0.05). There were no significant differences found among the body weights of YSBL female adults fed on wheat, rice, and sweet maize (wheat: 0.473 ± 0.028 g versus rice: 0.462 ± 0.029 g versus sweet maize: 0.249 ± 0.025 g) (*p* < 0.05).

## 4. Discussion

The YSBL has historically been regarded as the second worst pest species affecting bamboo forestry operations [7]. Migratory pests may feed on different host plants [29], especially during long-distance migration. *Schistocerca gregaria* (Forskål) and *Locusta migratoria manilensis* (Meyen) have a diverse host-plant range and impact on agriculture, forestry, and animal husbandry. Similarly, the YSBL also has a comparable living habitat to both locust species [6,30]. The suitability of plants in a landing region can highly influence their survival. This study found that YSBL nymphs preferred to feed on wheat and rice rather than waxy maize and sweet maize, and a temperature of 30 °C was suitable for their survival and development under laboratory conditions. 

Insects are ectothermic and heterothermic organisms, in contrast to endothermic and homeothermic mammals. The body temperature of most insects is linked to changes in ambient temperature and highly influences their development and their metabolic and physiological rates [31]. Hence, different insects have different optimal temperatures. Larvae of *Galleria mellonella* L. normally develop at a high constant temperature of about 30 °C in the beehive [32]. *Anthonomus grandis grandis* Boheman prefers mean temperatures between 20 °C and 30 °C [33]. This implies that the YSBL may find sufficient acceptable food sources beyond the bamboo forests of Yunnan Province, China. Thus, it is vital to take the necessary measures to control this pest in the earlier stages before it spreads into a larger region.

This study found that the preference of YSBL larvae for the tested hosts is not invariable at different instars. First instar nymphs prefer maize, rice, and waxy maize to sweet maize; the mature nymphs and adults prefer wheat and rice to waxy maize and sweet maize. Previously, studies have found that phytophagous insects have evolved a physiological regulatory digestive mechanism through co-evolution with plants in order to achieve optimal adaptation to diverse host plants [21]. The host preference variation in different instars of YSBL nymphs may be attributed to their physiological regulatory digestive mechanism. Moreover, despite the selective attraction of insects to their native hosts, populations of phytophagous insects may be compelled to exploit alternative host plants. These divergences can arise due to various factors, such as the scarcity of preferred host resources in their environment or a subset of individuals within the population developing a preference for a novel host that offers greater utilization efficiency compared to the original one [34,35]. Although the developmental duration of the 1st instar was observed to be significantly shorter on waxy maize than sweet maize, some individual YSBL nymphs exhibited no ecdysis and remained alive for an extended period of time before dying at 35 °C, particularly 2nd and 3rd instar nymphs fed on waxy maize seedlings and 4th instar nymphs fed on sweet maize. The reason could be that the different nutrient components of maize leaves may vary during their growth. The sucrose content in maize leaves first decreases and then increases during maize growth [36]. In the average feeding area, the 1st instar YSBL nymphs preferred maize to *Phyllostachys edulis* (Carrière) J. Houz. and *Miscanthus sinensis* Andersson, while the 2nd to the 5th instars of YSBL nymphs probably preferred the latter two to maize [37]. For *S. frugiperda*, the earlier instar larvae fed on cotton leaves, and later instars preferred squares and bolls [38]. However, the finding of host preferences only reflects laboratory conditions, which do not imitate the circumstances of migration and regional invasion.

Remarkably, China is considered to be the largest wheat producer globally, and the production of wheat ranks third after maize and rice [39]. At present, damage by the YSBL on wheat has not been reported in the field. We reported favorable survival rates for the YSBL on wheat and rice in the laboratory. In addition to exploring host preferences, we showed that 30 °C is the optimal temperature for the YSBL’s growth. The effect of host plants and temperature on the bionomics of the YSBL has strongly manifested itself in the life table parameters and in phenotype characteristics, especially survival rates, developmental duration, and body length. Indeed, the YSBL can complete full development and an entire lifecycle within the range of 25 °C to 30 °C depending on the availability of host plants. It is unclear whether the YSBL has different crop preferences as it develops, and it would therefore be of value to carry out more fundamental research to understand the causes and mechanisms of its development. 

## 5. Conclusions

In conclusion, the life table parameters and phenotype characteristics of the YSBL were used to analyse the suitability of different hosts under different temperatures; the results showed that the host plant and temperature had a significant effect on the biological parameters of the YSBL. The findings of the present study will contribute to understanding the roles of the host plant and temperature on the development, survival, and suitability of the YSBL. This study provides basic information for further research on whether the YSBL’s preference for different host plants is caused by plant-specific nutrients or plant characteristics. In addition, our results could be used for the early warning of wheat and rice damage in Yunnan Province and other regions.

## Figures and Tables

**Figure 1 biology-12-01377-f001:**
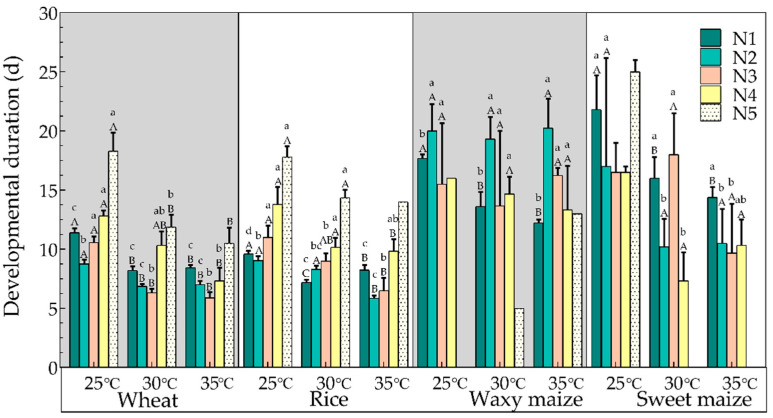
Mean developmental duration of YSBL nymphs fed on different hosts at different temperatures. The lowercase letters (a–c) indicate significant differences in the developmental duration of YSBL nymphs on different hosts at the same temperature. The capital letters (A–C) indicate significant differences in the developmental duration of YSBL nymphs at different temperatures on the same host. An absence of letters above the bar means that there were fewer than two observations. Note: N1–N5 represent first, second, third, fourth, and fifth instars, respectively.

**Figure 2 biology-12-01377-f002:**
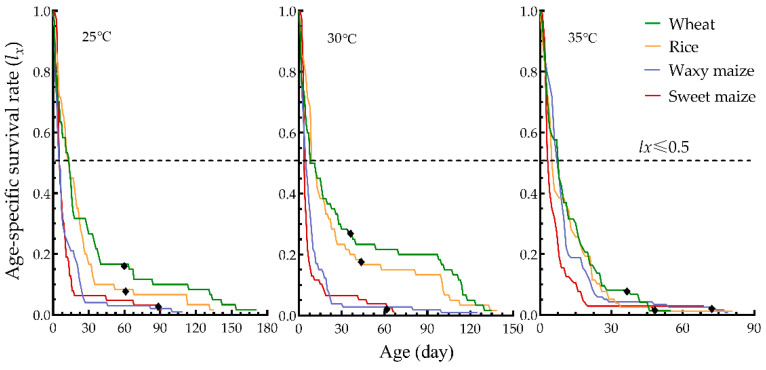
Age-stage specific survival rate (*l_x_*) of YSBLs fed on different hosts and at different temperatures. Note: the black rhombus means the emergence date of the first adult.

**Figure 3 biology-12-01377-f003:**
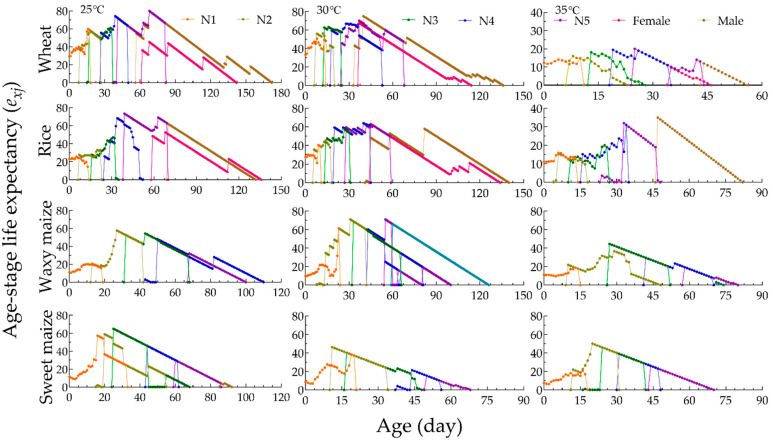
Age-stage specific life expectancy (*e_xj_*) of YSBLs fed on different hosts at different temperatures. Note: N1–N5 represent first, second, third, fourth, and fifth instars, respectively.

**Figure 4 biology-12-01377-f004:**
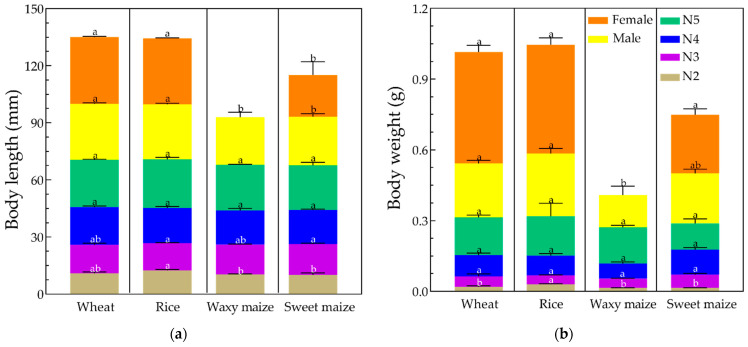
Mean body length (**a**) and body weight (**b**) of YSBLs fed on different hosts. The lowercase letters (a and b) indicate significant differences in the body length and body weight on different hosts at the same developmental stage. Note: N2–N5 represent second, third, fourth, and fifth instar, respectively.

**Table 1 biology-12-01377-t001:** Regression models between developmental rates and temperatures.

Developmental Stage	Host	Regression Quadratic Model	*R* ^2^	F
N1	Wheat	*V* = −0.600 + 0.045 × *T* − 0.001 × *T*^2^	0.744	8.711 *
	Rice	*V* = −0.916 + 0.068 × *T* − 0.001 × *T*^2^	0.651	5.597 *
N2	Wheat	*V* = −0.541 + 0.043 × *T* − 0.001 × *T*^2^	0.670	6.104 *
	Rice	*V* = 0.633 − 0.040 × *T* + 0.001 × *T*^2^	0.809	12.678 **
N3	Wheat	*V* = −1.122 + 0.078 × *T* − 0.001 × *T*^2^	0.794	9.633 *
N5	Wheat	*V* = −0.722 + 0.050 × *T* − 0.001 × *T*^2^	0.727	6.649 *
	Rice	*V* = −0.239 + 0.019 × *T* − 0.001 × *T*^2^	0.883	11.328 *

*V*: developmental rate. *T*: temperature. *R*^2^: correlation coefficient. * *p* < 0.05, ** *p* < 0.01. Note: N1–N3 and N5 represent first, second, third, and fifth instars, respectively.

**Table 2 biology-12-01377-t002:** Mean (± SE) adult emergence rate of YSBLs fed on four hosts at different temperatures.

Temperature (°C)	Wheat	Rice	Waxy Maize	Sweet Maize
25	0.17 ± 0.07 a A	0.08 ± 0.06 a AB	-	0.03 ± 0.02 a
30	0.25 ± 0.06 a A	0.18 ± 0.04 a A	0.02 ± 0.02 b A	-
35	0.07 ± 0.03 a A	0.02 ± 0.02 a B	0.02 ± 0.02 a A	-

The lowercase letters (a and b) indicate significant differences in the adult emergence rate of YSBLs on different hosts at the same temperature. The capital letters (A and B) indicate significant differences in the adult emergence rate of YSBLs at different temperatures on the same host. Note: - means no adult emergence under the corresponding treatment.

## Data Availability

The datasets generated and/or analyzed in the current study are available from the corresponding author upon reasonable request.

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
