# Peer review of "An Evaluation of the Crop Preference and Phenotypic Characteristics of Ceracris kiangsu Tsai (Orthoptera: Arcypteridae) under Different Temperatures"

_biology, 2023, doi:10.3390/biology12111377_

Round 1

Reviewer 1 Report

Comments and Suggestions for Authors

Reviewer’s comments

I have carefully reviewed your manuscript titled "The host preference and phenotypic characteristics of Ceracris kiangsu Tsai (Orthoptera: Arcypteridae) under different temperatures" and appreciate the effort you have put into this research. Your study offers valuable insights into the host preference and phenotypic characteristics of the yellow-spined bamboo locust (YSBL), which can significantly impact agricultural regions.

Overall, your research is promising, but there are some areas that require attention and clarification:

1.      Please supplement some specific numerical results in the abstract, such as survival rates and developmental times, to provide a more comprehensive summary of the findings.

2.      For the introduction, consider introducing key concepts like life table parameters and fitness characteristics earlier to better prepare readers for the subsequent sections. Additionally, citing relevant studies on host preference and temperature effects in other pest species would strengthen your argument.

3.      In the Results, please provide more comprehensive explanations for the observed differences and trends in the data. Help readers understand why certain instars developed more quickly on specific host plants or at particular temperatures. Including statistical values (e.g., p-values) would support the significance of your findings.

4.      Discussion, consider comparing your findings to previous research on YSBL or similar pests. This can provide context and highlight the novelty of your work. Additionally, elaborate on the practical implications of your results for pest management strategies, particularly how they might be applied to protect wheat and rice crops.

5.      Acknowledge any limitations of your study, such as the laboratory setting. Suggest areas for future research that could address these limitations and further advance our understanding of YSBL behavior.

Please address these minor queries as well:

1.      Could you provide more details about the specific varieties or cultivars of wheat, rice, waxy maize, and sweet maize used in your experiments? Information about their origin or source and any unique characteristics could be insightful.

2.      Why the temperature of 30 °C was chosen as the focus of your study? How does this temperature relate to the natural conditions in the regions affected by YSBL?

3.      The study suggests that YSBL nymphs' preference for hosts varied at different instars. What might explain this variation? Are there specific nutrients or plant characteristics that influenced their choice?

4.      How the findings from your laboratory experiments might relate to the migratory behavior of YSBL in the field? How might the availability of host plants during migration affect their survival and damage potential?

5.      Practically, how might your research findings be applied to protect wheat and rice crops from YSBL damage? Are there specific strategies or interventions that your results suggest?

I did not identify any major redaction or drafting errors in the manuscript. However, there are a few areas where improvements can be made for clarity and precision:

1.      In the abstract: The phrase "since 2014, there had many outbreaks of YSBL in Laos" seems grammatically incorrect. It could be rephrased for clarity.

2.      In the introduction: The sentence "For phytophagous insects, the plant hosts are the fundamental condition to support their survival and may have extensive effects on the fitness, fecundity, and reproductive strategies of insects" could be rephrased for smoother readability.

3.      In discussion: The sentence "This study found that the preference of YSBL larvae for the tested hosts is not invariable at different instars" could be clarified for better comprehension.

4.     In conclusion: In the sentence, "The findings of the present study will contribute to understanding the roles of host plant and temperature on the development, survival, and suitability of YSBL," it might be helpful to specify how these findings contribute to practical applications or future research.

5.      These are relatively minor issues, and overall, the manuscript is well-structured and scientifically sound. Your study has the potential to make a significant contribution to the field of pest management, especially in regions where YSBL poses a threat to staple crops. Addressing these suggestions and queries will enhance the clarity and impact of your research. I look forward to seeing your revised manuscript and commend your dedication to advancing our understanding of YSBL behavior.

regards, 

The other reviewer.

Author Response

I have carefully reviewed your manuscript titled "The host preference and phenotypic characteristics of Ceracris kiangsu Tsai (Orthoptera: Arcypteridae) under different temperatures" and appreciate the effort you have put into this research. Your study offers valuable insights into the host preference and phenotypic characteristics of the yellow-spined bamboo locust (YSBL), which can significantly impact agricultural regions.

Overall, your research is promising, but there are some areas that require attention and clarification:

  1. Please supplement some specific numerical results in the abstract, such as survival rates and developmental times, to provide a more comprehensive summary of the findings.

Authors’ reply: These sentences were modified: "The developmental durations of 1st and 3rd instar YSBL on wheat (1st: 8.21 ± 0.35 d; 3rd: 6.32 ± 0.34 d) and rice (1st: 7.19 ± 0.23 d; 3rd: 9.00 ± 0.66 d) were significantly shorter than that on waxy maize (1st: 13.62 ± 1.22 d; 3rd: 13.67 ± 6.33 d) and sweet maize (1st: 16.00 ± 1.79 d; 3rd: 18.00 ± 3.49 d) at 30 ËšC. Body lengths of male and female YSBL on wheat (male: 29.52 ± 0.40 mm, female: 34.97 ± 0.45 mm) and rice (male: 28.85 ± 0.68 mm, female: 34.66 ± 0.35 mm) were significantly longer than when fed on sweet maize (male: 25.64 ± 1.60 mm, female: 21.93 ± 6.89 mm)."

  1. For the introduction, consider introducing key concepts like life table parameters and fitness characteristics earlier to better prepare readers for the subsequent sections. Additionally, citing relevant studies on host preference and temperature effects in other pest species would strengthen your argument.

Authors’ reply: we add more information as "Life table and its parameters, such as developmental duration, age-specific survival rate, and age-stage specific life expectancy can be used to better understand and analyze insect population dynamics, make predictions and develop integrated pest management plans [19]. Body length of Aeolothrips intermedius Bagnall was the most responsive phenotype characteristics to host plant [20]. Body weight of Hyphantria cunea (Drury) feeding on the high-preference host plants were significantly higher than those feeding on the low-preference host plants [21]."

  1. In the Results, please provide more comprehensive explanations for the observed differences and trends in the data. Help readers understand why certain instars developed more quickly on specific host plants or at particular temperatures. Including statistical values (e.g., p-values) would support the significance of your findings.

Authors’ reply: More extra information provided as "Insects are ectothermic and heterothermic organisms, in contrast to endothermic and homeothermic mammals. The body temperature of most insects is linked to changes in ambient temperature, and highly influence their development, metabolic and physiological rates [32]."

  1. Discussion, consider comparing your findings to previous research on YSBL or similar pests. This can provide context and highlight the novelty of your work. Additionally, elaborate on the practical implications of your results for pest management strategies, particularly how they might be applied to protect wheat and rice crops.

Authors’ reply: It is revised as "Schistocerca gregaria (Forskål) and Locusta migratoria manilensis (Meyen) had a diverse host plant range and impact on agriculture, forestry, and animal husbandry, Similarly YSBL also has comparable living habitat to both locust species [6,31]."

Wheat, rice, and maize are cultivated in entire China including Yunnan Provice. Our laboratory findings indicate that YSBL may all the tested crop plants, and more prefer wheat.

  1. Acknowledge any limitations of your study, such as the laboratory setting. Suggest areas for future research that could address these limitations and further advance our understanding of YSBL behavior.

Authors’ reply: This sentence was added: "This study provides basic information for further researches on whether YSBL's preference for different host plants is caused by plant-specific nutrients or plant characteristics. "

We also mentioned this in the Discussion: "Previously studies found that phytophagous insects have evolved a digestive physiological regulatory mechanism through co-evolution with plants in order to achieve optimal adaptation to diverse host plants [21]...Besides, despite the selective attraction of insects to their native hosts, populations of phytophagous insects may be compelled to exploit alternative host plants."

Please address these minor queries as well:

  1. Could you provide more details about the specific varieties or cultivars of wheat, rice, waxy maize, and sweet maize used in your experiments? Information about their origin or source and any unique characteristics could be insightful.

Authors’ reply: More details provided as "For this study, we chose four different staple crops in China including common wheat (Aikang 58, Henan Bainong seed industry Co., LTD), rice (Xiangzaoxian 45, Hunan Dongting Gaoke seed industry Co., LTD), waxy maize (Jinghuangnuo, Beijing Huaao Nongke Jade breeding development Co., LTD) and sweet maize (Suketian 1506, Nanjing Jiahua Agricultural development Co., LTD), which were disease-resistant or widely cultivated in China [23–26]."

  1. Why the temperature of 30°C was chosen as the focus of your study? How does this temperature relate to the natural conditions in the regions affected by YSBL?

Authors’ reply: We revised them. "The body temperature of most insects is linked to changes in ambient temperature, and highly influence their development, metabolic and physiological rates [32]. Hence, different insects have different optimal temperatures. Larvae of Galleria mellonella L. normally develop at high constant temperature about 30 ËšC in the beehive [33]. Anthonomus grandis grandis Boheman preferred the mean temperatures between 20 ËšC to 30 ËšC [34]."

  1. The study suggests that YSBL nymphs' preference for hosts varied at different instars. What might explain this variation? Are there specific nutrients or plant characteristics that influenced their choice?

Authors’ reply: More information provided as "Previously studies found that phytophagous insects have evolved a digestive physiological regulatory mechanism through co-evolution with plants in order to achieve optimal adaptation to diverse host plants [21]. The host preference variation in different instars of YSBL nymphs may be attributed to their digestive physiological regulatory mechanism."

Besides, the sucrose content in maize leaves exhibited a pattern during the growth stage, initially declining and subsequently increasing, this factor also may have contributed to the aforementioned phenomenon. These views have also been reflected in the "Discussion".

  1. How the findings from your laboratory experiments might relate to the migratory behavior of YSBL in the field? How might the availability of host plants during migration affect their survival and damage potential?

Authors’ reply: Once YSBL migrated and landed in the new region, YSBL adults may cause the damage to wheat, rice and maize if they were cultivated. In case, the YSBL may die due to limited and little host plants. Meanwhile, the rich of host plants will provide enough nutrient for adults and their oviposition as well as the offspring population.

  1. Practically, how might your research findings be applied to protect wheat and rice crops from YSBL damage? Are there specific strategies or interventions that your results suggest?

Authors’ reply: One side, wheat and rice are cultivated in Yunnan Province, which is a significant agricultural region in China. The life cycle of YSBL, including both nymph and adult stages, closely resembles that of crops. Furthermore, the adult insects lay eggs subsequent to feeding, potentially exerting direct impacts on the following year's crop yield.

On the other side, although there are currently no specific protection strategies in place, implementing trapping methods farmland could be considered as a potential approach to mitigate the damage caused to wheat and rice crops in the field.

I did not identify any major redaction or drafting errors in the manuscript. However, there are a few areas where improvements can be made for clarity and precision:

  1. In the abstract: The phrase "since 2014, there had many outbreaks of YSBL in Laos" seems grammatically incorrect. It could be rephrased for clarity.

Authors’ reply: Thanks for your comments. The sentence was modified: "Since 2014, there have been many outbreaks of YSBL populations in Laos."

  1. In the introduction: The sentence "For phytophagous insects, the plant hosts are the fundamental condition to support their survival and may have extensive effects on the fitness, fecundity, and reproductive strategies of insects" could be rephrased for smoother readability.

Authors’ reply: Thanks for your comments. It was revised as "For phytophagous insects, the plant hosts represent a fundamental condition that support their survival and may have extensive effects on the fitness, fecundity, and reproductive strategies of insects [1,2]."

  1. In discussion: The sentence "This study found that the preference of YSBL larvae for the tested hosts is not invariable at different instars" could be clarified for better comprehension.

Authors’ reply: Thanks for your comments. It was revised as " 1st instar nymphs are preferring maize, rice and waxy maize than sweet maize, the mature nymphs and adults prefer the wheat and rice than waxy maize and sweet maize.”

  1. In conclusion: In the sentence, "The findings of the present study will contribute to understanding the roles of host plant and temperature on the development, survival, and suitability of YSBL," it might be helpful to specify how these findings contribute to practical applications or future research.

Authors’ reply: Thanks for your comments. It was revised as "This study provides basic information for further researches on whether YSBL's preference for different host plants is caused by plant-specific nutrients or plant characteristics."

In addition, we have a forward-looking outlook in theDiscussion: "It is unclear that YSBL have different crop preference with the development, thus it is value to carry out more fundamental research to understand the causes and mechanisms."

These are relatively minor issues, and overall, the manuscript is well-structured and scientifically sound. Your study has the potential to make a significant contribution to the field of pest management, especially in regions where YSBL poses a threat to staple crops. Addressing these suggestions and queries will enhance the clarity and impact of your research. I look forward to seeing your revised manuscript and commend your dedication to advancing our understanding of YSBL behavior.

Authors’ reply: thanks a lot for your review and useful comments. We did the improvements carefully.

Reviewer 2 Report

Comments and Suggestions for Authors

The paper deals with a potential harmful pest to some staple crops in China, studying the influence of temperature and host species on important vital parameters, such as developmental time,  survival rates and body mass. It would have been very interesting to add one more piece of information about fecundity as well: this would have permitted to calculate an index like rm which could aid to predict if a population is expected to increase or decline under particuar condition, but of course to get a wider range of data  is not always possible, and this is a good starting point. Overall the paper is satisfactory about methodology and analysis, even if I have some concern about some missing data (see fig 4) and the presentation of some data (You often state that some  data are different: I think you should put less emphasis on these differences, when they are not significative). In the discussion you should explain more clearly how your data could help monitoring the pest: e.g. based on data of bamboo forest distribution respect to cultivated area and in relation to migratory capacity, the capacity of this host to be an important source  to staple crops. In the complex the paper provide interesting data and deserves to be published

Comments on the Quality of English Language

Quality of Englis is below standard. I made some corrections, but above all discussion and conclusions must be extensively rwritten and  assistance of a native speaker is reccomended

Author Response

  1. Actually the tests of the paper could be more convincingly considered as suitability

Authors’ reply: after rethinking, the title has been revised as “Evaluate the crop preference and phenotypic characteristics of Ceracris kiangsu Tsai (Orthoptera: Arcypteridae) under different temperatures”.

  1. Actually in the paper there is no evidence of damage-- In short, this study implied that YSBL is more harmful to wheat and rice than maize.

Authors’ reply: we revised it as “this study implied that YSBL prefer to wheat and rice.

  1. There were about 200 ha forest land found invaded YSBL from Laos in 2023. Where?

Authors’ reply: extra information provided as “in Yunnan Province, China”.

  1. I don't think this heading is meaningful I would merge 2.1 and 2.2

Authors’ reply: 2.1 and 2.2 were merged as one part “2.1 Insect culture”. More details see the revised version.

  1. Most of the differences are not significative I think this paragraph should be organized in a different way, such as: even though developmental rate on wheat was higher for all tested temperature respect to the other crop, the only significative contrat was observed for...

Authors’ reply: This has been revised. The adult emergence rate of YSBL fed on the wheat seedlings was the highest, which was higher than that on the other three crops at 25 ËšC, 30 ËšC, and 35 ËšC (Table 2). The adult emergence rate of YSBL fed on wheat seedlings was significantly higher than that fed on waxy maize at 30 ËšC (P < 0.05).

The highest adult emergence rates of YSBL fed on wheat and rice seedlings were observed at 30 ËšC, which were higher than at 25 ËšC and 35 ËšC (Table 2). The adult emergence rate of YSBL fed on rice at 30 ËšC was significantly higher than that at 35 ËšC (P < 0.05).

  1. You could use an acronym to define this parameter and use it later when you mention it for 30 and 35 °C

Authors’ reply: The first age (FA) means that survival rate of the day was not more than 0.5 (i.e., lx ≤ 0.5).

  1. You should say somewhere that on waxy maize you didn't obtain any female.

Authors’ reply: Added in.

  1. Unclear sentences in discussion have been revised. More extra information has been provided.
